# The Pharmacokinetics of Medetomidine Administered Subcutaneously during Isoflurane Anaesthesia in Sprague-Dawley Rats

**DOI:** 10.3390/ani10061050

**Published:** 2020-06-18

**Authors:** Leila T. Kint, Bhedita J. Seewoo, Timothy H. Hyndman, Michael W. Clarke, Scott H. Edwards, Jennifer Rodger, Kirk W. Feindel, Gabrielle C. Musk

**Affiliations:** 1Faculty of Health and Medical Sciences, The University of Western Australia, Perth 6009, Australia; leila.kint@uwa.edu.au; 2Centre for Microscopy, Characterisation and Analysis, Research Infrastructure Centres, The University of Western Australia, Perth 6009, Australia; bhedita.seewoo@research.uwa.edu.au (B.J.S.); Kwfeinde@dal.ca (K.W.F.); 3Experimental and Regenerative Neurosciences, School of Biological Sciences, The University of Western Australia, Perth 6009, Australia; jennifer.rodger@uwa.edu.au; 4Brain Plasticity Group, Perron Institute for Neurological and Translational Science, Perth 6009, Australia; 5School of Veterinary Medicine, Murdoch University, Perth 6150, Australia; t.hyndman@murdoch.edu.au; 6Metabolomics Australia, Centre for Microscopy, Characterisation and Analysis, The University of Western Australia, Perth 6009, Australia; michael.clarke@uwa.edu.au; 7School of Animal Veterinary Sciences, Charles Sturt University, Wagga Wagga 2650, Australia; sedwards@csu.edu.au; 8School of Biomedical Sciences, the University of Western Australia, Perth 6009, Australia; 9Animal Care Services, the University of Western Australia, Perth 6009, Australia

**Keywords:** functional MRI, rat anaesthesia, refinement

## Abstract

**Simple Summary:**

Rodents, including rats, are used as animal models for research investigating neurological diseases in humans. To enable this research the animals are anaesthetized to facilitate imaging of the brain, but the anaesthetic drugs impact the results of the research. To minimize the variation between studies anaesthetic protocols should be similar. A common anaesthetic regime is the combination of two drugs (medetomidine and isoflurane); however, there is much variation in the doses of these drugs and the way in which they are administered. To provide some evidence to facilitate the standardization of anaesthetic protocols this study was performed to elucidate the details of what the body does to these drugs when they are administered in a certain way. Three groups of rats were studied to determine the desired dose of medetomidine when isoflurane is used at a low dose (approximately 0.5%). The results of the study are an evidence-based suggestion for medetomidine and isoflurane anaesthesia during functional magnetic resonance imaging (fMRI) studies.

**Abstract:**

Anaesthetic protocols involving the combined use of a sedative agent, medetomidine, and an anaesthetic agent, isoflurane, are increasingly being used in functional magnetic resonance imaging (fMRI) studies of the rodent brain. Despite the popularity of this combination, a standardised protocol for the combined use of medetomidine and isoflurane has not been established, resulting in inconsistencies in the reported use of these drugs. This study investigated the pharmacokinetic detail required to standardise the use of medetomidine and isoflurane in rat brain fMRI studies. Using mass spectrometry, serum concentrations of medetomidine were determined in Sprague-Dawley rats during medetomidine and isoflurane anaesthesia. The serum concentration of medetomidine for administration with 0.5% (vapouriser setting) isoflurane was found to be 14.4 ng/mL (±3.0 ng/mL). The data suggests that a steady state serum concentration of medetomidine when administered with 0.5% (vapouriser setting) isoflurane can be achieved with an initial subcutaneous (SC) dose of 0.12 mg/kg of medetomidine followed by a 0.08 mg/kg/h SC infusion of medetomidine. Consideration of these results for future studies will facilitate standardisation of medetomidine and isoflurane anaesthetic protocols during fMRI data acquisition.

## 1. Introduction

Anaesthetic protocols using a combination of medetomidine and isoflurane, are increasingly being used in functional magnetic resonance imaging (fMRI) studies of the rodent brain [1,2,3,4,5,6,7,8,9,10]. The use of medetomidine for these studies was first reported in 2002 [9], whilst the use of low dose isoflurane (<0.5% vapouriser setting) in conjunction with medetomidine as an anaesthetic regime was first reported in 2012 [3,4,8].

Medetomidine is an α_2_-adrenoceptor agonist that causes sedation, hypertension, bradycardia, respiratory depression, hyperglycaemia, diuresis, muscle relaxation and analgesia [11,12,13,14,15,16,17,18,19,20,21,22,23,24,25,26]. The potency and receptor selectivity of medetomidine has led to its widespread use in veterinary anaesthesia, mostly in dogs and cats [27]. Medetomidine causes sedation through the activation of central α_2_-adrenoceptors in the locus coeruleus, which prevents excitatory neurotransmitter release in the central nervous system and thereby depresses cortical arousal [13,14,15]. Vascular side effects of medetomidine occur due to the activation of peripheral α_2_-adrenoceptors, which causes a transient and marked increase in systemic vascular resistance [17,18]. This vasoconstriction is followed by a decrease in vascular tone due to suppression of central nervous system-mediated sympathetic stimulation on blood vessels.

Isoflurane is a GABAergic fluorinated ether that causes anaesthesia, respiratory depression, bronchodilation, vasodilation, hypotension and muscle relaxation [28,29]. Isoflurane is commonly used for clinical and veterinary anaesthesia due to its rapid onset of action, short recovery time, safety and titratability [29,30,31]. The minimum alveolar concentration of isoflurane in adult Sprague-Dawley rats is 1.46 ± 0.06% [32].

The benefit of combining medetomidine with isoflurane specifically for fMRI studies has been described. When >0.1% isoflurane is administered with medetomidine, the epileptic activity caused by medetomidine is suppressed [4,21,22]. Furthermore, the drug combination allows for maintenance of a steady state of anaesthesia for >4 h, with consistent fMRI data [3]. In contrast, when medetomidine is administered alone via a constant rate infusion (subcutaneous (SC) or intravenous (IV)), it is not possible to maintain a steady state of sedation for >3 h. It has been reported that medetomidine administered alone can only be used in fMRI experiments >3 h if the initial infusion dose is increased three-fold after 90 min, or if medetomidine is specifically administered using an initial IV injection of at least 0.05 mg/kg medetomidine followed by a subsequent continuous SC or IV infusion of at least 0.1 mg/kg/h medetomidine, whereby the initial dose cannot be omitted, and the dose cannot be decreased [1,33].

Despite the increasing popularity of this combination of medetomidine and isoflurane, a standardised anaesthetic protocol for their combined use for rodent brain fMRI studies has not been established [4,34]. Various protocols are described with variable doses of both medetomidine and isoflurane, different routes of administration of medetomidine, and variation in the time of fMRI data collection relative to the time of medetomidine administration [3,4,6,7,34,35,36]. For example, in seven rodent fMRI studies employing medetomidine and isoflurane anaesthesia, the dose of isoflurane for maintenance of anaesthesia varied from 0.25–1.4% [3,4]. Furthermore, reported loading doses for medetomidine range from 0.03 to 0.15 mg/kg and the subsequent infusion doses range from 0.03 to 0.1 mg/kg/h [35,36]. In addition, the initial injection was administered via the intravenous (IV), intramuscular, intraperitoneal or subcutaneous (SC) routes and the infusion via the IV, intramuscular or SC routes. The time of fMRI data collection after the initial administration of medetomidine ranged from 15 min to 90 min [3,7,36]. This variation in the use of medetomidine and isoflurane in rodent brain fMRI studies may be attributed to a lack of comprehensive data on the pharmacokinetics and pharmacodynamics of medetomidine in rodents. Importantly, the serum concentration of medetomidine when administered with low dose isoflurane for rodent brain fMRI studies is unknown. Thus, the rationale for the administration of medetomidine alongside isoflurane for rodent brain fMRI studies is largely derived empirically [1,2,3,4,5,6,7,8,9]. However, there is now evidence that both resting-state and evoked blood-oxygen-level-dependent (BOLD) fMRI signals are altered by the type of anaesthetic drug(s) used and their dose [37,38,39,40,41,42,43,44]. Thus, the many aforementioned inconsistencies in the use of medetomidine and isoflurane may be hindering the interpretation, generalisation, meta-analysis and reproducibility of rodent brain fMRI studies.

Medetomidine substantially reduces the dose of isoflurane required to achieve stable anaesthesia, therefore minimising anaesthetic-induced distortions of BOLD fMRI signals. When dogs are administered a dose of 0.03 mg/kg IV medetomidine, there is a reduction of the minimum alveolar concentration of isoflurane by 47.2% [45]. Furthermore, when rodents are anaesthetised with a combined medetomidine and isoflurane dose of 0.06 mg/kg/h IV and 0.5–0.6%, respectively, they exhibit levels of anaesthesia comparable to rodents treated either medetomidine 0.1 mg/kg/h IV or isoflurane 1.3% [34]. Reducing the dose requirement of each drug is beneficial, as high doses of each drug in isolation are associated with significant drug-specific distortions of BOLD fMRI signals [34]. This artefact occurs because BOLD fMRI studies rely on the coupling between local blood flow and local neuronal activity (known as neurovascular coupling) to infer and therefore measure neural activity [44,46,47]. BOLD signals in anaesthetised rodents are considered an accurate measure of neural activity when they produce an image reflective of brain activity in the awake rodent. Conversely, BOLD signals are considered inaccurate when they produce an image reflective of fMRI-induced-stress or anaesthetic-induced changes in the BOLD effect [34,48]. Recent evidence suggests that BOLD fMRI signals obtained during medetomidine and isoflurane anaesthesia can be used to accurately measure rodent brain activity [34]. This attribute can be partially explained by the synergistic effects of the drugs on preserving neurovascular coupling [3,35]. When administered alone, medetomidine alters the BOLD effect by causing cerebral vasoconstriction, bradycardia, decreased cerebral blood flow and altered astrocyte activity [4,35,49]. In contrast, when isoflurane is administered alone, it alters the BOLD effect by inducing vasodilation in cerebral vasculature [1,50]. Accordingly, when medetomidine and isoflurane are administered together, medetomidine appears to attenuate isoflurane-induced cerebral vasodilation, leading to better preservation of neurovascular coupling [51].

To better utilise medetomidine and isoflurane anaesthesia in rodent fMRI studies, their use should be standardised. To this end, the pharmacokinetic profile of medetomidine during combined medetomidine and isoflurane anaesthesia needs to be elucidated, and the serum concentration of medetomidine in this context needs to be identified. The aim of this study was to describe the pharmacokinetics of medetomidine during isoflurane anaesthesia and determine the serum concentration of medetomidine when administered with 0.5% (vapouriser setting) isoflurane, so that an evidence-based dosing regimen of medetomidine could be determined for rat brain fMRI studies.

## 2. Materials and Methods

The study was approved by the University of Western Australia’s Animal Ethics Committee (RA/3/100/1599) and conducted in accordance with the Australian code for the care and use of animals for scientific purposes, 8th edition [52]. The rats were housed in an AAALAC (Association for the Assessment and Accreditation of Laboratory Animal Care) facility.

### 2.1. Animals

Twenty-four male, eight-week-old, Sprague-Dawley rats (*Rattus norvegicus*) were imported from the Animal Resources Centre (Canning Vale, WA, Australia) as specific pathogen free rats. Rats were transported in groups to the animal care facility and held for at least three days prior to the study. The rats were housed in a temperature-controlled environment on a 12 h light-dark cycle with food and water ad libitum at M-block in QEII Medical Centre (Nedlands, WA, Australia). The cages were individually ventilated with minimum dimensions of 38.8 cm wide, 40.6 cm long and 21 cm high on coarse aspen bedding. The rats were housed in pairs, fed a commercial rat diet (Specialty Feeds Meat Free Rat and Mouse Diet, Glen Forrest, Australia) that was autoclaved prior to introduction into the animal facility and were provided with acidified drinking water (pH 2.5–3). Food was not withheld prior to anaesthesia. On the day of the procedure, the rats were transferred to the Centre for Microscopy Characterisation and Analysis (University of Western Australia, Nedlands, Australia).

### 2.2. Experimental Procedure

The rats were randomly allocated to three experimental groups: Group T for determination of the target serum concentration of medetomidine when administered with low dose isoflurane for rodent brain fMRI studies (*n* = 8); Groups IV and SC for determination of the SC bioavailability of medetomidine during isoflurane anaesthesia (*n* = 8 each).

On the days of the procedures, the rats were anaesthetised with isoflurane (Isothesia™, Henry Schein Animal Health, 2000, Australia) in an induction chamber (4% isoflurane in 100% medical oxygen, 2 L/min). Once adequately anaesthetised (recumbent, no response to toe pinch) the rats were transferred onto the experimental benchtop and positioned for delivery of isoflurane throughout the experiment (0.5–2% isoflurane vapouriser setting in 100% medical oxygen, 1.5 L/min, Darvall Zero Dead Space face mask circuit, Advanced Anaesthesia Specialists) under a heat lamp. Physiological monitoring included body temperature, respiratory rate, heart rate, electrocardiography (PC-SAM Small Animal Monitor, SA Instruments Inc., 1030 System), exhaled isoflurane and CO_2_ (data not shown) (ISATM Sidestream Gas Analyzer, Masimo Sweden AB and PHASEIN and Lightning Multi-Parameter Monitor Vetronic Services Ltd., Newton Abbot, UK) and blood glucose concentration (Accu-Chek Guide, Roche, Mannheim, Germany). These variables were recorded every 5 min. A single rat was studied at any one time, during the hours of 8 a.m. and 6 p.m.

Medetomidine (1 mg/mL, Ilium Medetomidine Injection, Troy Laboratories Pty. Limited, Glendenning, Australia) was administered according to the treatment group. In Group T, rats were administered an initial dose of medetomidine of 0.05 mg/kg SC over 1 s via a 29 G insulin syringe (BD Ultra-Fine Insulin Syringe, Becton Dickinson Pty Ltd., Macquarie University Research Park North Ryde, Australia), immediately followed by a continuous medetomidine infusion of 0.15 mg/kg/h SC, administered via a 25 G butterfly catheter connected to a single syringe infusion pump (Legato 100 Syringe Pump, KD Scientific Inc., Holliston, MA, USA). This protocol was developed empirically and used in our laboratory [10]. In the IV and SC groups, rats were manually administered a single dose of either IV (through a catheter placed in a lateral tail vein) or SC (under the skin over a flank) medetomidine at 0.05 mg/kg. The concentration of isoflurane was immediately reduced to 0.5% after administration of the initial dose of medetomidine and then subsequently altered to maintain an adequate depth of anaesthesia as assessed by response to toe pinch, heart rate and respiratory rate.

For serial blood sampling, a catheter was placed in the lateral tail vein (22 G, 1 IN, BD Angiocath IV Catheter, BD Australia, Seven Hills, NSW, Australia), secured with surgical tape and flushed with heparinised saline (5 IU/mL). In Group T, blood samples were collected 60 and 90 min after the initial dose of medetomidine. The conditions during anaesthesia were consistent with those observed in previous studies performed in this laboratory and were considered suitable for identification of the target concentration of medetomidine. In the IV group, blood was collected before medetomidine administration and 2, 5, 10, 20, 30, 60, 120 and 180 min afterwards. In the SC group, blood was collected before medetomidine administration and 10, 20, 30, 40, 50, 60, 120, 180 and 240 min afterwards. Following collection of the final sample, but before recovery from anaesthesia, the rats were euthanised via an intraperitoneal or IV injection of pentobarbitone (160 mg/kg, Lethabarb, Jurox, Rutherford, Australia).

### 2.3. Blood Sampling

Approximately 0.5 mL of blood was collected at each timepoint by inserting a 23 G butterfly catheter (SV*23BLK, Terumo Australia Pty Ltd., Macquarie Park, NSW, Australia) into the injection port of the tail vein catheter. The initial saline-diluted drops of blood were discarded before sample collection. A glucometer was used to immediately measure the blood glucose concentration (Accu-Chek Guide, Roche, BellaVista, Australia). After each sample, the catheter was flushed with 0.5 mL of heparinised saline (5 IU/mL) to prevent clot formation in the catheter and replace blood volume. In the event that sufficient blood could not be collected from the catheter, blood was drawn percutaneously from the lateral saphenous veins, medial saphenous veins or femoral arteries through a butterfly catheter.

All blood samples were collected in 3 mL Eppendorf tubes and allowed to clot at room temperature for 10 min before refrigeration. Refrigerated samples were centrifuged within 4 h of collection using an Eppendorf MiniSpin plus centrifugation at 2000× *g* for 10 min. Approximately 0.2 mL of serum supernatant from each sample was collected and transferred into new 3 mL Eppendorf tubes. These serum samples were then frozen at −80 °C.

### 2.4. Serum Analysis

The analyses were performed at Metabolomics Australia (University of Western Australia, Nedlands, Australia). Medetomidine concentrations of the serum samples were analysed using a liquid chromatography-tandem mass spectroscopy (LC-MS/MS) technique. The internal standard during analysis was medetomidine-13C,d_3_ hydrochloride (Sapphire Bioscience, Redfern, Australia).

To process the serum for analysis, 20 µL of serum were added to 50 µL of working internal standard (50 ng/mL labelled medetomidine-13C,d_3_ in 50:50 methanol:water plus 0.1% formic acid) and vortexed for 10 s. The mixture was then vortexed with 1 mL ethyl acetate for 120 s, after which they were centrifuged at 3000 rpm for 5 min. Then, 900 µL solvent were evaporated to dryness for 30 min at 40 °C before being reconstituted in 70 µL of 50:50 methanol:water.

Processed serum extracts of 2 µL were run on an Agilent 6460 LC-MS/MS in 2D mode using isotope dilution to adjust for instrument response. Solvent A was LC-MS/MS grade water (Thermo Optima) with 0.1% formic acid (Merck). Solvent B was LC-MS/MS grade methanol (B & J) and 0.1% formic acid (Merck). Column one was an Agilent 2.1 × 50 mm 2.6 µm C18 Poroschell and column two was a Phenomenex Kinetex 3 × 150 mm 2.6 µm Biphenyl phase. The flow rate was set at 0.5 mL/min and a gradient was run from 50% B to 80% B in 10 min. The column was washed with 98% B and then returned to 50% B by 7 min. Compounds were heart cut from column one to column two between 0.4–0.9 min. Medetomidine and medetomidine-13C,d_3_ were monitored with transitions 201 > 95 and 204.1 > 98, respectively, with a collision energy of 15. Assay calibration was achieved by spiking drug free matrix matched rat plasma to create a calibration curve, with the r^2^ typically >0.9999. Assay precision was assessed during the project by extracting 4 samples in triplicate and the intra-assay CV ranged from 2.1–5.7%. The limit of quantitation for the assay was 0.1 ng/mL.

### 2.5. Pharmacokinetic and Pharmacodynamic Calculations

The maximum serum concentration (*C*_max_) of medetomidine following SC administration was the highest measured concentration for each animal. The time at *C*_max_ (*t*_max_) was also determined. The elimination rate constant (*λ_z_*) was calculated as the negative slope of the semilogarithmic plot of each animal created from the terminal three time points (t = 120, 180 and 240 min). The elimination half-life (*t_1/2β_*) was calculated as ln(2)/*λ_z_*. The area under the serum concentration time curve (*AUC*_0__→__∞_) was estimated by the trapezoidal rule extrapolated to infinite time. Standard formulae were used to calculate the total body clearance (*Cl* = dose/*AUC*) [53] and volume of distribution at pseudo-equilibrium (*Vd_area_* = *Cl*/*λ_z_*) [54].

The target serum concentration of medetomidine (*C_target_*) was obtained from the rats in Group T and was taken as the mean serum concentration of MED at t = 60 and 90 min. The loading dose (LD) was estimated from the product of *Vd_area_* and *C_target_*. The maintenance dose rate (MD) was calculated from the product of *Cl* and *C_target_*.

### 2.6. Trial of Results

To trial the calculated drug administration regime for SC administration of medetomidine an additional two rats were administered medetomidine with isoflurane to ensure the conditions for anaesthesia were stable and uneventful. The dose of medetomidine in these two trials was an initial SC dose of 0.12 mg/kg medetomidine delivered over 5 s followed by a SC infusion of 0.08 mg/kg/h with 0.5% (vapouriser setting) isoflurane.

### 2.7. Statistical Analyses

Data were tested for normality using a D’Agostino and Pearson test and compared using Student’s *t*-test or Mann–Whitney test (GraphPad Prism). The *p*-value used to define statistical significance was 0.05. Data are expressed as mean ± standard deviation or as otherwise stated.

## 3. Results

### 3.1. Group T

The rats weighed 333.2 ± 19.3 g (*n* = 6). Data from two rats were excluded from the study due to inaccurate weight records at the time of anaesthesia and therefore incorrect doses of medetomidine being administered. Otherwise anaesthesia was uneventful and a stable heart rate (307.9 ± 30.7 bpm), respiratory rate (52.9 ± 8.3 breaths/min) and normothermic temperature (38.1 ± 0.7 °C) were maintained. The blood glucose concentration at 60 min was 20.9 ± 3.0 mmol/L and at 90 min was 23.2 ± 2.6 mmol/L (Figure 1). The vapouriser setting for inhaled isoflurane was maintained at 0.5% after induction of anaesthesia, whereby from 5 to 90 min after the initial dose of medetomidine the exhaled isoflurane concentration was 0.49 ± 0.05% (Table 1).

The serum medetomidine concentration at 60 min after the initial medetomidine dose was 13.9 ± 3.9 ng/mL (range 9.9–20.8 ng/mL) which was similar to that at 90 min (*p* = 0.329): 15.0 ± 2.0 ng/mL (range 12.0–18.1 mg/mL). Therefore, for the purposes of identifying the serum concentration of medetomidine when administered with low dose isoflurane for rat brain fMRI studies, these data were grouped, and the target serum concentration of medetomidine was determined to be 14.4 ± 3.0 ng/mL.

### 3.2. Group IV

The rats weighed 333.6 ± 17.2 g (*n* = 8). In seven of the Group IV rats, respiratory arrest was observed immediately after manual administration of the IV injection of medetomidine, and gentle external chest compressions were performed. After 2 min, spontaneous ventilation resumed. Anaesthesia was otherwise uneventful. The blood glucose concentration peaked at 60 min at 16.6 ± 2.4 mmol/L and at 180 min at 11.2 ± 2.3 mmol/L.

Fifteen minutes after the administration of IV medetomidine, half of the rats required the vapouriser setting for isoflurane to be increased from 0.5% isoflurane. By 35 min, all the rats required the vapouriser setting for isoflurane to be increased from 0.5% isoflurane and maintained at approximately 1–2%.

The serum medetomidine concentration peaked at 2 min at 754.6 ± 672.5 ng/mL (range 122.1–2139.4 ng/mL). Given the variability of these data, the IV group was excluded from pharmacokinetic calculations.

### 3.3. Group SC

The rats weighed 317.9 ± 19.9 g (*n* = 7). Data from one rat (the first) was excluded from the study as it was administered an initial SC dose of medetomidine of 0.1 mg/kg and became apnoeic for approximately 2 min, requiring external chest compressions. The seven subsequent rats were administered a lower dose of 0.05 mg/kg SC medetomidine and anaesthesia was uneventful. The blood glucose concentration peaked at 120 min at 20.4 ± 3.4 mmol/L (Figure 1).

The inhaled isoflurane concentration required to maintain an adequate depth of anaesthesia throughout the procedure in Group SC was more variable than in the other groups (Table 1). The serum medetomidine concentration peaked at 60 min at 3.4 ± 0.9 ng/mL (Figure 2).

### 3.4. Pharmacokinetic Calculations

The pharmacokinetic and pharmacodynamic parameters were calculated from the mean serum medetomidine concentration data (Table 2). To achieve a target medetomidine concentration of 14.4 ± 3.0 ng/mL an initial SC dose of 0.12 mg/kg medetomidine followed by a SC infusion of 0.08 mg/kg/h medetomidine should be administered during isoflurane anaesthesia.

### 3.5. Trial of Results

Two additional rats were administered medetomidine with isoflurane at the doses calculated in this study. The vapouriser setting for isoflurane could be maintained at or below 0.5% and anaesthesia was uneventful. Given the calculated initial dose was higher than that used in groups SC and IV the initial dose was administered over five seconds to mitigate the risk of apnoea (as observed in the IV group and the first rat in the SC group that was administered 1.0 mg/kg SC). Apnoea did not occur when the initial dose was delivered over five seconds.

## 4. Discussion

The present study shows that steady state serum concentrations of medetomidine will be achieved in male Sprague-Dawley rats if an initial SC dose of medetomidine of 0.12 mg/kg is administered in combination with continuous 0.5% (vapouriser setting) isoflurane, followed by a SC infusion of medetomidine at 0.08 mg/kg/h. This regime appears to provide suitable conditions for anaesthesia when the initial dose is delivered over five seconds. This result is within the range of doses reported in the literature [34,35].

The Group T result was used as the target serum concentration of medetomidine when administered with 0.5% (vapouriser setting) isoflurane. The anaesthetic protocol in this group was based on consultation with researchers using combined medetomidine and isoflurane anaesthesia in ongoing resting-state rodent fMRI studies. Given the apparent empirical success of the protocol in achieving strong and reproducible fMRI signals [10], rats under this protocol were hypothesised to achieve a steady state concentration of medetomidine. Data from rats in Group SC were used to determine the SC bioavailability of medetomidine during combined medetomidine and isoflurane anaesthesia. Collectively, the data from the two groups were used to inform the SC administration of medetomidine in rodents with low dose isoflurane.

The intention was to use data from both the IV and SC groups to perform pharmacokinetic calculations. However, the data from Group IV were excluded from the analysis due to considerable variation in this data set. We attribute the variation to the use of a single cannula for both IV drug administration and subsequent serial blood sampling. Issues arising from the use of a single cannula have been investigated and described by Gaud et al. [55]. They report that the use of a single cannula is not suitable for pharmacokinetic studies. Some compounds will experience non-specific binding to the cannula that may contaminate the first few blood samples taken from the cannula and lead to overestimation of serum concentrations [55]. Manually flushing the cannula with heparinised saline can help dislodge bound medetomidine, therefore reducing serum concentration overestimation. However, the flushing can also cause increased variation in measured serum medetomidine concentration due to the random error associated with repeated hand-operated techniques. This oversight likely led to inaccurate serum concentrations in the Group IV and hence exclusion of these data.

The Group T data suggest that a steady state serum medetomidine concentration of 14.4 ± 3.0 ng/mL is suitable for rats undergoing brain fMRI with 0.5% (vapouriser setting) isoflurane. This combination of drugs creates conditions suitable for prolonged anaesthesia (hours) without major anaesthetic-specific distortion of BOLD fMRI signals [10]. Future studies could consider using various doses of medetomidine and isoflurane to better define the therapeutic range for these drugs in the context of optimising the quality of fMRI images.

In the present study, the elimination half-life of medetomidine in rats was calculated to be 65.2 (±9.0) min. Similar values were calculated by Bol et al. (56.2 and 57.4 min) in a study of dexmedetomidine that was administered to Harlan-Sprague-Dawley male rats by two different IV infusion protocols [11]. In our study, and the work by Bol et al., drug concentrations were analysed for 210–240 min, and in neither study did blood medetomidine nor dexmedetomidine concentrations become undetectable. In contrast, a slower elimination half-life (1.6 h) was reported by Salonen et al. after tritium (3H)-labelled medetomidine was administered SC to male and female Sprague-Dawley rats. Furthermore, Salonen et al. detected plasma radioactivity at five and eight hours after administration of 3H- medetomidine [56]. The persistence of medetomidine at these time points (five and eight hours) may suggest that in our four-hour study, and in the study by Bol et al., the elimination rate constant was overestimated and therefore the elimination half-life was underestimated. This parameter could be explored in future studies by quantifying blood medetomidine (or dexmedetomidine) levels for several hours after administration.

The first rat in Group SC was administered a rapid initial dose (<one second, delivered manually) of 0.1 mg/kg SC medetomidine and became apnoeic for approximately two minutes. This response was not previously observed when the initial SC medetomidine dose was mechanically delivered in one second. Thus, the decision was made to alter the initial SC dose in Group SC from 0.1 mg/kg to 0.05 mg/kg for the remaining seven rats in that group. Although rats in Group IV also became apnoeic after administration of medetomidine, the dose in this group was not altered. The rationale to not alter the dose in Group IV was that transient apnoea could be managed with manual external chest compressions with the rat in sternal recumbency. To mitigate the risk of apnoea, the initial dose could be delivered over a longer time period; so during the trial of the calculated initial and infusion dose, the initial dose was administered over five seconds by the infusion pump. The conditions during anaesthesia were stable and uneventful.

Measurement of blood glucose concentrations was performed opportunistically and was not the primary aim of the project. Nevertheless, hyperglycaemia developed in all the rats in this study and although this side effect of medetomidine is described in rats its impact on experimental outcomes is not clear [57]. The mechanism of hyperglycaemia is a combination of anti-ADH (antidiuretic hormone) effects and alterations in insulin sensitivity, resulting in an osmotic diuresis [58]. This side effect of administration of medetomidine should be considered when designing anaesthetic regimens for research.

There are a number of limitations to this study which must be considered when interpreting the results. Only male, eight-week-old, Sprague-Dawley rats were used in this small study. This cohort limits the direct applicability of the results to female rats, other rat strains and mice. The age of the rats in this study is also a limitation of the model as adult animals may have a different pharmacokinetic profile for medetomidine. Future studies could expand the applicability of these results by investigating the pharmacokinetics of medetomidine in female rats, pregnant rats, obese rats, different ages and strains of rats and mice. In addition, the pharmacokinetic calculations could only be performed with serum concentrations of medetomidine that were obtained following SC administration. The data from Group IV was unfortunately excluded. Nevertheless, the data from Group SC were utilized in isolation, which meant that during the sampling period, the serum concentrations of medetomidine were assumed to be in pseudo-equilibrium. Thus, calculating the volume of distribution using the area method (*Vd_area_*) was appropriate [59]. The loading dose should be calculated using the volume of distribution calculated at steady state (*Vd*_ss_). Given *Vd_area_* is usually only larger than *Vd*_ss_ by a small amount, our calculated loading dose is likely to still be a reliable estimate. Furthermore, single doses of medetomidine and isoflurane were evaluated in this study as the aim was to determine a target concentration of medetomidine based upon empirical evidence of using these doses. Future work should consider the evaluation of alternative doses and their impact on fMRI outputs. Finally, the target dose as determined by Group T was based on the premise that quality fMRI images were acquired (in previous work in the lab) with the empirical protocol. Correlation of our conclusions with the quality of fMRI images has not been performed.

For studies where multiple imaging sessions are scheduled and the animals recover from anaesthesia, the administration of atipamezole is prudent. This drug antagonises medetomidine and is routinely administered in the laboratory in which this study was performed when rats recover from anaesthesia.

The benefit of combined medetomidine and isoflurane anaesthetic protocols in rodent brain fMRI studies may be compromised by inconsistencies in these anaesthetic protocols between studies. Anaesthetics alter BOLD fMRI signals and these inconsistencies hinder the interpretation, generalisation, meta-analysis and reproducibility of rodent brain fMRI studies. Future brain fMRI studies should consider an evidence-based approach to the use of medetomidine and isoflurane anaesthetic protocols to standardise the regime between studies.

## 5. Conclusions

The data suggest that a serum medetomidine concentration of 14.4 ± 3.0 ng/mL is suitable for rats undergoing brain fMRI with 0.5% (vapouriser setting) isoflurane.

## Figures and Tables

**Figure 1 animals-10-01050-f001:**
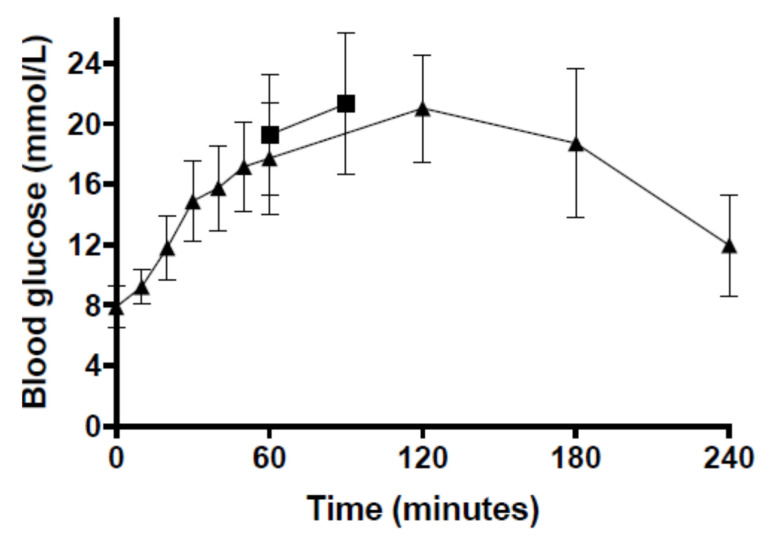
Time course of mean (±standard deviation) blood glucose concentration during anaesthesia of Sprague-Dawley rats in Group T (0.05 mg/kg medetomidine subcutaneous (SC) followed by a continuous infusion of 0.15 mg/kg/h SC with 0.5% isoflurane; squares) and Group SC (0.05 mg/kg medetomidine SC; triangles).

**Figure 2 animals-10-01050-f002:**
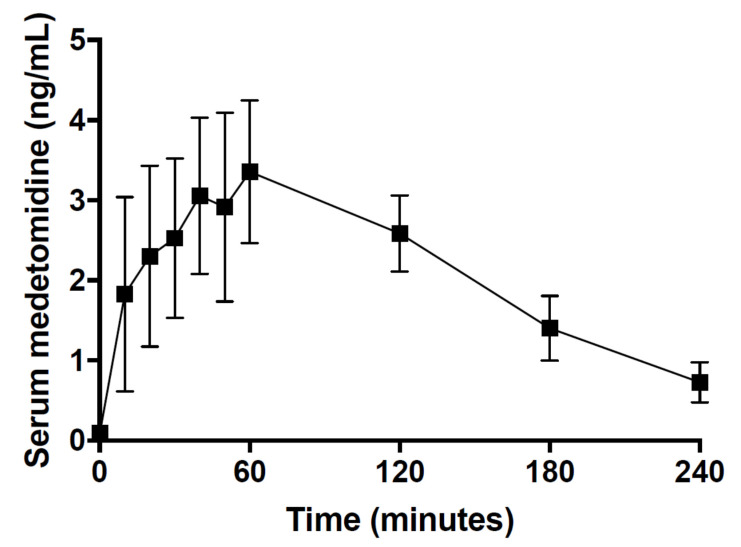
Time course of mean (±standard deviation) serum medetomidine concentration after subcutaneous administration of 0.05 mg/kg medetomidine in six Sprague-Dawley rats.

**Table 1 animals-10-01050-t001:** Mean (±standard deviation) concentration of expired isoflurane during administration of isoflurane after an initial dose of medetomidine of 0.05 mg/kg SC followed by a continuous medetomidine infusion of 0.15 mg/kg/h SC (Group T) or a medetomidine dose of 0.05 mg/kg SC (Group SC). The delivery of isoflurane was adjusted as necessary to maintain an adequate depth of anaesthesia, as assessed by response to toe pinch, heart rate, respiratory rate and expired carbon dioxide concentration. Only results for the first 90 min are shown.

Expired Isoflurane (%)								
	5 min	10 min	15 min	25 min	35 min	45 min	60 min	90 min
Group T (*n* = 6)	0.6 (± 0.2)	0.6 (± 0.4)	0.5 (± 0.3)	0.5 (± 0.04)	0.5 (± 0.1)	0.5 (± 0.1)	0.4 (± 0.1)	0.5 (± 0.1)
Group SC (*n* = 7)	1.2 (± 0.5)	1.0 (± 0.5)	1.0 (± 0.3)	1.0 (± 0.1)	1.0 (± 0.1)	1.0 (± 0.1)	1.0 (± 0.2)	1.2 (± 0.3)

**Table 2 animals-10-01050-t002:** Individual and mean (±standard deviation) pharmacokinetic (PK) and pharmacodynamic parameters after administration of 0.05 mg/kg SC medetomidine in seven Sprague-Dawley rats. *C_max_* = maximum serum concentration; *t_max_* = time of *C_max_*; *λ*_z_ = elimination rate constant; *t_1/2β_* = elimination half-life; *AUC_0→_**_∞_* = area under the serum concentration time curve from time = 0 to ∞; *Cl* = total body clearance; *Vd_area_* = volume of distribution at pseudo-equilibrium; LD = loading dose; MD = maintenance dose.

Rat ID	*C_max_* (ng/mL)	*t_max_* (min)	*λ*_z_ (/min)	*t_1/2β_* (min)	*AUC_0→_**_∞_* (ng.min/mL)	*Cl* (mL/kg/min)	*Vd_area_* (L/kg)	LD (mg/kg)	MD (mg/kg/h)
R	4.9	60	0.0095	73.0	664.7	75.2	7.9	0.1142	0.0651
S	3.3	60	0.0118	58.7	570.8	87.6	7.4	0.1070	0.0758
T	4.4	50	0.0112	61.9	610.1	82.0	7.3	0.1055	0.0709
U	3.7	40	0.0130	53.3	485.3	103.0	7.9	0.1143	0.0891
V	2.9	120	0.0107	64.8	511.8	97.7	9.1	0.1316	0.0845
W	2.8	60	0.0112	61.9	485.6	103.0	9.2	0.1325	0.0891
X	3.3	120	0.0084	82.5	704.4	71.0	8.5	0.1218	0.0614
Mean (SD)	3.6 (0.7)	72.9 (30.6)	0.0108 (0.0014)	65.2 (9.0)	576.1 (81.1)	88.5 (12.1)	8.2 (0.7)	0.1181 (0.0101)	0.0765 (0.0105)

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
