# Peer review of "The Pharmacokinetics of Medetomidine Administered Subcutaneously during Isoflurane Anaesthesia in Sprague-Dawley Rats"

_animals, 2020, doi:10.3390/ani10061050_

Round 1

Reviewer 1 Report

Please refer to attached file.

Author Response

General comments:

  • The introduction would benefit from re-ordering the paragraphs. It would be more logical for the authors to start by explaining the rationale for using the combination of low dose isoflurane with medetomidine for rodent fMRI, and then move on to discussing problems of using varying doses and routes of administration.

AU: The introduction has been modified as suggested.

  • The description of the aim and conclusions of the study is misleading as fMRI quality (and so therapeutic effect) was not actually assessed in this study – comments below.

AU: These concerns are addressed as requested.

Specific comments: Line 39 – 41: re-phrase to make it clear that the authors did not assess therapeutic effects in this study. Please specify what 0.5% isoflurane means – is this the expired concentration?

AU: This issue has been addressed in the abstract and introduction.

Line 51: Typographical error ‘20029’

AU: This error has been corrected.

Lines 52 (and again frequently throughout the text: lines 74, 91, 104, 161, 178): What does the % dose of isoflurane represent? Is this the vaporiser setting, inspired concentration, end-tidal concentration? This needs to be specified each time isoflurane ‘dose’ is referred to.

AU: We have added ‘vapouriser setting’ when the dose of isoflurane is referred to.

Lines 74 – 76: This sentence could be re-phrased to make it more concise; it seems unnecessarily complicated. E.g. “reported loading doses for medetomidine in rodent fMRI range from X – X, and infusion rates from X-X.”

AU: This sentence has been reworded as suggested.

Line 124 – 126: This sentence is misleading. It suggests that the authors are going to identify the therapeutic concentration of medetomidine in the context of fMRI. In fact, they identified the serum concentration that corresponded to previously reported IV/SC administered doses that provided a clinically desirable effect.

AU: We have removed the word therapeutic throughout the manuscript.

Lines 126 – 130: I understand what the authors are trying to explain here but it is confusing and on first read this sentence does not fit or make sense. I think instead, it must be made clear that the doses investigated here are based on those that have previously shown to produce clinically desirable effects in this setting. Similar to what is stated in lines 326 – 334 of the discussion. The sentence as it is currently written infers that assessment of fMRI quality was performed in this study.

AU: We have removed this sentence.

Lines 130 – 133: Again, it is misleading to state that the authors would “determine the therapeutic serum concentration”. Rather they determined the serum concentration of medetomidine when administered at doses previously shown to have the desired therapeutic effects.

AU: We have removed the word therapeutic in the manuscript.

 Lines 153 – 155: “when administered with low dose isoflurane for rodent brain fMRI studies”. This suggests that fMRI was carried out in this study, when it does not appear to have been. This requires re-phrasing as mentioned above. 2

AU: We have removed the word therapeutic.

Lines 170 – 175: consider adding rationale for the doses chosen here, rather than leaving this until the discussion (lines 326 – 334). I would consider the methods used to determine the medetomidine dose for Group T part of the methodology.

AU: A statement adding the rationale has been included as suggested.

Lines 320-322: Again, re-phrasing is required to make it clear that the authors determined the serum concentration of medetomidine when administered at doses previously shown to have the desired therapeutic effects, rather than making their own assessment of therapeutic effects during this study.

AU: This sentence has been reworded as suggested.

Lines 415 – 416 (conclusion). Please re-phrase. This is not a representative conclusion of the study performed here as the therapeutic effects of during fMRI were not assessed.

AU: This sentence has been reworded as suggested.

Reviewer 2 Report

Major concerns

  • 8w old animals were used, this is rather young. How can this affect the outcome. Are the results found extrapolatable towards adult full grown animals.
  • A combination of bolus/infusion with isoflurane would have an added value.

Major advices

  • The manuscript would profit from adition of a schematic overview of the groups and what was administered when and when the samples were taken.
  • When isoflurane had to be increased over time, how was this evaluated?

Minor comments

  • check line 51 for a typing mistake in the date.
  • check line 168 for phrasing

Author Response

  • 8w old animals were used, this is rather young. How can this affect the outcome. Are the results found extrapolatable towards adult full grown animals.

AU: This issue has been acknowledged in the limitations of the study, as suggested.

  • A combination of bolus/infusion with isoflurane would have an added value.

AU: We’re not sure what you mean. In this study a bolus loading dose of medetomidine followed by an infusion is described.

Major advices

  • The manuscript would profit from addition of a schematic overview of the groups and what was administered when and when the samples were taken.

AU: We haven’t added a schematic overview as suggested as neither of the other 2 reviewers thought this would benefit the manuscript.

  • When isoflurane had to be increased over time, how was this evaluated?

AU: This information was originally included: The vapouriser setting for isoflurane was immediately reduced to 0.5% after administration of the initial dose of medetomidine and then subsequently altered to maintain an adequate depth of anaesthesia as assessed by response to toe pinch, heart rate and respiratory rate.

Minor comments

  • check line 51 for a typing mistake in the date.

AU: This error has been corrected.

  • check line 168 for phrasing

AU: This sentence has been revised for clarity.

Reviewer 3 Report

This is an interesting paper and raises many issues for further research.  Its salvation is that it comes with some practical anaesthetic regimes to try on other strains and female rats to produce acceptable fMRIs with a more stable evoked blood-oxygen-level-dependent (BOLD) fMRI signal.  The authors quite rightly and comprehensively point out the technical and practical limitations of their work and I commend them on elucidating those caveats.

They use a metal needle tail vein cannula which has the disadvantage of raising more injury to the endothelium than a polythene cannula would have done, which could be important if serials fMRIs are anticipated.  They might also like to consider using the jugular as opposed to the caudal/tail vein.

Minor comments

I would like to authors to check the following typos please.

Line 51: 20029 seems a long way off ?  Seems to refer to reference 9 which itself is incomplete.  I assume it is 2002 with an inadvertent bracket?

Line 140 should be Rattus norvegicus, not norwegicus

Lines 202 et seq:   All blood samples were collected in 3 mL Eppendorf tubes and allowed to clot at room etc.  It is unlikely to be able to obtain 0.2ml serum from 0.5ml blood.  It is likely to be a plasma/serum mix?  Why not just collect into an appropriate anticoagulant and use plasma and spin immediately after collection

Line 217-218:  Processed serum extracts of 2 μL were run on an Agilent 6460 LC-MS/MS in 2D mode using isotype dilution to adjust for instrument response. Solvent A was LC-MS/MS grade water (Thermo ….        Please check isotype and not isotope?

The summary emphasises 0.5% isoflurane but actually 1% is finally used?

Author Response

This is an interesting paper and raises many issues for further research.  Its salvation is that it comes with some practical anaesthetic regimes to try on other strains and female rats to produce acceptable fMRIs with a more stable evoked blood-oxygen-level-dependent (BOLD) fMRI signal.  The authors quite rightly and comprehensively point out the technical and practical limitations of their work and I commend them on elucidating those caveats.

AU: Thank you for your kind words.

They use a metal needle tail vein cannula which has the disadvantage of raising more injury to the endothelium than a polythene cannula would have done, which could be important if serials fMRIs are anticipated.  They might also like to consider using the jugular as opposed to the caudal/tail vein.

AU: In fact, we used a polythene cannula and appreciate the comment about using the jugular vein for future studies.

Minor comments

I would like to authors to check the following typos please.

Line 51: 20029 seems a long way off ?  Seems to refer to reference 9 which itself is incomplete.  I assume it is 2002 with an inadvertent bracket?

AU: This error has been corrected.

Line 140 should be Rattus norvegicus, not norwegicus

AU: This error has been corrected.

Lines 202 et seq:   All blood samples were collected in 3 mL Eppendorf tubes and allowed to clot at room etc.  It is unlikely to be able to obtain 0.2ml serum from 0.5ml blood.  It is likely to be a plasma/serum mix?  Why not just collect into an appropriate anticoagulant and use plasma and spin immediately after collection

AU: Serum was the sample type preferred by the laboratory for LC-MS.  We did manage to harvest close to 0.2 mL of serum from our samples.

Line 217-218:  Processed serum extracts of 2 μL were run on an Agilent 6460 LC-MS/MS in 2D mode using isotype dilution to adjust for instrument response. Solvent A was LC-MS/MS grade water (Thermo ….        Please check isotype and not isotope?

AU: This error has been corrected – to isotope.

The summary emphasises 0.5% isoflurane but actually 1% is finally used?

AU: In order to keep the rats asleep in groups SC and IV the dose of isoflurane had to be increased to more than 0.5%.  The pharmacokinetic calculations are based on the combination of medetomidine with no more than 0.5% isoflurane.  We have ensured that this meaning is clear.